# Effect of Low-CTE Oxide-Dispersion-Strengthened Bond Coats on Columnar-Structured YSZ Coatings

Christoph Vorkötter [1],*  , Daniel Emil Mack [1]  , Dapeng Zhou [1]  , Olivier Guillon [1,2] and Robert Vaßen [1]

1   Forschungszentrum Jülich GmbH, Institute of Energy and Climate Research, Materials Synthesis and Processing (IEK-1), 52425 Jülich, Germany; d.e.mack@fz-juelich.de (D.E.M.); d.zhou@fz-juelich.de (D.Z.); o.guillon@fz-juelich.de (O.G.); r.vassen@fz-juelich.de (R.V.)
2   Jülich Aachen Research Alliance, JARA-Energy, 52056 Aachen, Germany
*   Correspondence: c.vorkoetter@fz-juelich.de; Tel.: +49-2461-61-9707

**Abstract:** Thermal barrier coatings (TBCs) are commonly used to protect gas turbine components from high temperatures and oxidation. Such coatings consist of ceramic top coats and metallic bond coats. The mismatch in thermal expansion of the top coat, the bond coat and the component material is one main factor leading to the failure of the coating system. Columnar-structured top coats offer an enhanced tolerance to the strain during thermal cycling. On a flat bond coated surface, these TBCs reach higher thermal cycling performance. However, on rough surfaces, as used for thermal spray coatings, the performance of these thermal barrier coatings seems to be restricted or even stays below the performance of atmospheric-plasma-sprayed (APS) thermal barrier coatings. This low performance is linked to out-of-plane stresses at the interface between the top coat and the bond coat. In this study, a thin additional oxide-dispersion-strengthened (ODS) bond coat with high alumina content provides a reduced mismatch of the coefficient of thermal expansion (CTE) between the top coat and the bond coat. Columnar suspension plasma sprayed (SPS), yttria-stabilized zirconia (YSZ) TBCs were combined with low-CTE ODS bond coats. The behavior of these TBCs was characterized with respect to thermal cycling performance and degradation in a burner-rig facility. The comparison showed an up-to-four-fold increase in the performance of the new system.

**Keywords:** oxide-dispersion strengthened; thermal barrier coating; columnar-structured

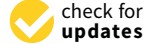



## 1. Introduction

Thermal barrier coatings (TBCs) are used to protect gas turbine components from high temperatures, oxidation and corrosion. A typical TBC consists of two layers: a porous ceramic top coat and a metallic bond coat. Both layers have different purposes and properties. The ceramic top coat provides thermal protection by offering a low thermal conductivity. The bond coat provides protection against oxidation and enhances the adhesion of the top coat. During thermal cycling, the different coefficients of thermal expansion of the top coat ($\alpha = 11 \times 10^{-6}$ K$^{-1}$) and the component material, for example Ni-based superalloys ($\alpha = 15$–$16 \times 10^{-6}$ K$^{-1}$), lead to stresses in the TBC system and ultimately to the failure of the coating system [1]. In addition, there are several other factors affecting the lifetime of the coating system. Another major factor is the oxidation of the bond coat [2,3], which introduces additional thermal stresses at the interface of the bond coat and the top coat due to the low coefficient of thermal expansion of the thermally grown oxide (TGO) [4,5]. Further factors are bond coat roughness influencing the adhesion of the top coat as well as the stress state, and the beta depletion (also named NiAl-phase depletion or aluminum depletion) of the bond coat leading to fast transient-oxide growth or a fracture in the toughness and porosity of the top coat [6]. Columnar-structured TBCs are more strain tolerant compared to conventional TBCs. The top coat can tolerate the tensile strain of the blade material during heating cycles by opening the spacings between columns.

Therefore, the performance of columnar-structured TBCs can exceed the performance of conventional TBCs [7,8].

Electron-beam physical-vapor deposition (EB-PVD) [9,10], or more recently, plasma-spray physical-vapor deposition (PS-PVD) are commonly used to spray columnar-structured thermal barrier coatings, while suspension plasma spraying is under development, especially to reduce production costs by avoidance of vacuum technology [11–14]. However, suspension plasma sprayed coatings on bond coats with a high roughness generally show rather poor thermal cycling performance [14,15]. This is contradictory to the findings in conventional atmospheric plasma sprayed (APS) TBCs, where enhanced adhesion of the top coat, due to higher roughness, results in higher thermal cycling performance [16]. To understand the failure mechanism of the columnar SPS coating, a detailed view of the deposition process and the effects on the interface during thermal cycling of these TBCs is necessary.

In the case of suspension plasma spraying, the formation of the columnar structure strongly depends on the undulations/roughness of the bond coat [11,13]. The deposition of columnar structures is a result of different impact angles of particles caused by the different inertia of the particles [17]. If the spray gun is aligned perpendicularly to the substrate, larger particles with a high momentum will perpendicularly impact the substrate. Fine particles with a low inertia follow the gas flow, which is redirected to a horizontal flow at the substrate surface. These fine particles will then impinge on asperities of the bond coat resulting in column deposition [17–19]. In a highly rough surface, the deposition of the columns is enhanced at the tips of a wavy bond coat, because already grown columns and roughness hills overshadow the deposition at the valleys. Due to this shadowing effect, the columns are surrounded by porosity. As the column adhesion to the bond coat is now mainly determined by the curved bond coat/top coat interface, stresses at the tip of this curvature, which are caused by the CTE mismatch between the bond coat and top coat during thermal cycling, can especially affect the adhesion of the coating [20].

During thermal cycling, the tensile in-plane stresses in TBC systems are relaxed at high temperatures after a rather short time. If such a system is cooled down, compressive in-plane and radial tensile stresses at the tips of the wavy bond coat/top coat interface will be introduced [2,4,20]. Cracks can grow under this condition. However, they are stopped at the valley locations, where there is compressive stress. Until a sufficiently thick TGO is formed during long-term operation, the resulting stress inversion leads to the extension of crack growth to the valley locations and ultimately to failure [21].

In the case of columnar-structured TBCs, the tensile stress at high temperatures and the amount of stress relaxation is reduced. This leads to lower in-plane compressive stresses at room temperature in the coating and hence, a reduced driving force for spallation [1].

The radial compressive stresses at the valleys of the wavy bond coat/top coat interface are reduced due to the porosity surrounding the columns [20]. With the lowered compressive stress at the valleys, cracks can easily propagate, leading to the premature failure of the coatings [20].

To take advantage of the high strain tolerance of the columnar-structured coatings and to achieve excellent thermal cycling performance, bond coats with low roughness can be used. Another means of achieving the high thermal cycling performance of columnar-structured TBCs is to apply a thin lamellar atmospheric plasma sprayed top coat at the interface of the top coat and the bond coat [12,20].

The scope of this work is to enhance the thermal cycling performance of SPS columnar TBCs by using a thin oxide-dispersion-strengthened (ODS) bond coat, which has already been successfully used with conventional thermal barrier coatings [2,16]. The ODS bond coat provides a reduced mismatch of the coefficient of thermal expansion between the metallic bond coat and the ceramic top coat. This can be expected to reduce the local stress levels at the curved interface structures. Moreover, the dispersed oxides can also enhance the oxidation resistance of the bond coat. With this combination, achieving high performance of a suspension plasma sprayed columnar top coat should be possible without the application of an additional interlayer of TBC by APS.

Columnar-structured yttria-stabilized zirconia TBCs with additional ODS bond coats of various alumina concentrations in the ODS material were applied to two (conventional) bond-coated types of Ni-based substrate materials. The thermal cycling performance and the failure mechanism of the TBC samples were studied by testing them until failure under thermal gradient conditions in a burner-rig facility. The performance of the TBCs was evaluated with respect to the bond coat temperature, failure position and coating parameters. The coating characteristics were determined by cross sections of the thermally cycled samples. The results are also compared in detail with data from columnar-structured TBCs without additional ODS bond coats, and with available data from conventional TBCs from previous studies at the same test conditions.

## 2. Materials and Methods

The raw material of oxide-dispersion-strengthened bond coats were produced by the high-energy milling of a powder mixture containing 10 wt.% and 30 wt.% alumina powder (Martoxid MR70, Martinswerk, Bergheim, Germany) and CoNiCrAlY bond coat powder (Amdry 995 Oerlicon Metco, Wholen, Switzerland). To achieve a powder-size distribution suitable for thermal spray, the powders were sieved to 20–56 µm. Details on processing, particle-size distributions, and chemical compositions were presented in [2]. As shown in previous studies, the deposition of the ODS bond coat requires only slight changes in the spray parameters to achieve a suitable deposition process [2,16].

**Sample manufacture:** The thermal barrier coatings were sprayed on button-type substrates made of Ni-based superalloys, namely the polycrystalline Inconel 738 (Doncaster Precision Castings-Bochum GmbH, Bochum, Germany) and the single-crystal superalloy ERBO 1, named and developed within the SFB Transregio 103 (chemical composition close to CMSX4, see [16]). The sample geometry (30 mm in diameter, 3 mm thickness and rounded edges) matched the thermal cycling burner-rig facility that was established in previous studies [22,23].

Vacuum plasma spraying (F4 gun Oerlikon Metco, Wohlen, Switzerland) was used to achieve highly dense coatings with a low oxide content. The bond coats were sprayed as a thin ODS bond coat (50 µm) on top of a conventional bond coat (120 µm). Bond-coated samples were heat treated for 4 h at 1140 °C and 16 h at 870 °C to increase the bond coat adhesion. Samples from [24] "Sample O" were specimens with a single-layered conventional bond coat (160 µm) on Inconel 738 and were used for reference, as produced by the same corresponding procedure. Details on those may also be found in a prior study [20]. The bond coat roughness, presented in Table A1 was measured by a Cyberscan CT350 with a confocal white-light sensor CHR1000 (CyberTECHNOLOGIES GmbH, Ingolstadt, Germany).

Sample names were derived from the first letters of the substrate material ("IN"-Inconel 738, "ER"-ERBO 1) followed by the alumina concentration in the upper bond coat ("10" or "30") or "0" for a single-layered conventional bond coat. Data on the samples with the conventional bond coat were taken from previous studies [11,24]. Where multiple samples were produced for the same ODS bond coat and substrate material combination, a capital letter was added to the sample name for enumeration.

For top coat deposition, a spray gun with axial suspension injection (Axial III Northwest Mettech Corporation, Vancouver, Canada) was used to apply the columnar YSZ top coat to all TBC thermal cycling samples. Spraying parameters are presented in Table 1. The suspension feeding rate was set to 29 g/min. For the suspension preparation, two YSZ powders TZ-5Y (9.7 wt.% yttria-stabilized zirconia powder Tosoh Corporation, Tokyo, Japan) and TZ-3Y (5.4 wt.% YSZ, Tosoh Corporation, Tokyo, Japan) were mechanically mixed with a ratio of 1:1 resulting in a 7.5 wt.% YSZ powder mixture. The ethanol-based suspension contained 900 g zirconia milling balls (3 mm diameter zirconia balls, Sigmund Lindner GmbH, Warmensteinnach, Germany), 700 g ethanol, 300 g powder mixture and 4.5 g dispersion agent (PEI, Ploysciences, Warrington, PA, USA). To achieve a homogeneous suspension with a limited amount of agglomerates, the suspension was milled by

low-energy milling at 120 rpm on a roller bench for at least 24 h. The milled suspension (30 wt.% YSZ solid load) was diluted to 5 wt.% YSZ solid load prior to spraying. Further information about the suspension can be found in [11].

**Table 1.** Process parameters for thermal spray.

| System | Torch Type | Torch Distance (mm) | Torch Speed (mm/s) | Current (A) | Voltage (V) | Ar/H$_2$ (N$_2$) (slpm) | Powder Feed Gas (slpm) | Feed Rate (g/min) | Pressure (mbar) |
|---|---|---|---|---|---|---|---|---|---|
| VPS | F4 | 275 | 440 | 640 /680 (ODS) | 70 | 50/9 (0) | 1.7 | 51–63 (Powder) | 60 |
| SPS [24] | Axial III | 70 | 1000 | 250 (3×) | 135 | 184/24 (37) | 15 | 30 (Suspension) | 1013 |

**Burner Rig Testing:** Thermal barrier coatings were thermally cycled in a burner-rig facility [22]. During the thermal cycling, the sample surfaces were heated by a flame from natural gas and oxygen to a temperature of 1300 °C while the backsides of the samples were cooled with pressurized air. Every 5 min, the surface-heating gas burner was removed and the surface was cooled by pressurized air for 2 min. The thermal gradient conditions were monitored by measuring the top coat-surface temperature with a pyrometer operating in the long-wavelength infrared range (KT15.85, Heitronics, Wiesbaden, Germany–spot size 12 mm) and the substrate temperature with a thermocouple positioned in the center of the substrate. An emissivity of 1 was used in the measurement of the surface temperature which, in the case of YSZ, led to only a slight underestimation of the temperature. The bond coat temperature was calculated from the thermal conductivities and nominal thicknesses of all layers combined with the surface and substrate temperatures at the steady-state condition by assuming linear temperature profiles throughout the thickness (1-dimensional Fourier's law). Based on this calculation, the surface and substrate temperatures were adjusted in order to achieve a bond coat temperature of about 1100 °C for all specimens. After thermal cycling tests, the effective bond coat temperature of each specimen was recalculated using the individual layer thicknesses derived from the specimen's cross section, as well as the averages of surface and bond coat temperatures recorded during the test [2].

Burner-rig tests were carried out in a fully automated facility that operated day and night, with the integrity of specimens checked periodically during working hours for macroscopic spalling or delamination of the coatings. In the case of temperature deviations of more than 50 K from the initial condition, the experiments were automatically stopped and it was determined whether or not the deviations could have been caused by malfunctions of the test equipment or if they were due to a critical degradation of the TBC system. A delamination of the coating will create an insulating air gap which is coupled with an increase in the surface temperature of the coating. A full spallation of the coating (locally) is coupled with a decrease in the surface temperature because the insulating layer that promotes the temperature drop has been (partially) lost. In principle, the test was terminated if delamination or spallation of the coating was observed exceeding an area of 25 mm$^2$ or if the desired temperature conditions could no longer be achieved in the test rig. The number of cycles at this condition was referred to as "cycles to failure". Damage to the coating within an area of 5 mm to the rim were neglected as those may have arisen from inhomogeneous load conditions at the rim.

**Microstructure Characterization:** The TBC sample cross sections were polished, platinum sputtered and analyzed by electron microscopy and optical microscopy. A Phenom (Phenomworld B.V., Eindhoven, Netherlands) and a Hitachi TM3000 (Hitachi High-Technologies Europe GmbH, Krefeld, Germany) electron microscope were used. The optical analysis was performed by a laser-scanning microscope Keyence VK-9710 (Neu-Isenburg, Germany). Layer thicknesses were determined by averaging measurement results from multiple SEM/laser-microscope images (for top coat and beta-depletion thickness, 5 laser

images; for TGO, bond coat and ODS-bond coat thickness, 10 SEM images). For each image the layer thicknesses were measured by Image J (National institute of health, Bethesda, MD, USA) software with 15 distance measurements per image. The standard deviation of the mean values for each image was taken as error. Colum density and vertical crack density were measured by counting the number of columns and the vertical cracks at half of the coating thickness in 7 laser-microscope images (each 1.4 mm wide) distributed over the whole sample. A column was defined as a small conical feature characterized by dark lines visible in SEM backscattered images. A dark vertical feature counts as a crack if it is perpendicular to the coating surface and its length is at least of 1/2 of the coating thickness.

## 3. Results and Discussion

For the samples analyzed within this study, some differences exist in the failure between the APS and columnar TBC samples. This is visible especially for conventional bond coat TBC systems Figure 1. As visible in Figure 1, the columnar-structured TBCs show buckling of the TBC, for example starting in the center of the coating, while the APS samples show failure often starting at the sample edge.

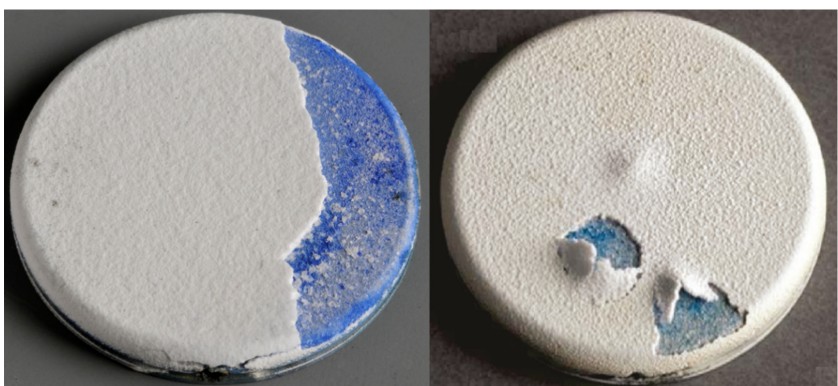

**Figure 1.** Top view of APS (**left**) and columnar SPS TBC (**right**) sample with yttria-stabilized zirconia top coat and conventional bond coat after test in the burner-rig facility, sample specification see Table A1 [2,24].

The thermal mismatch between the top coat and the bond coat is a main factor affecting the delamination of APS and columnar TBC samples. The higher thermal expansion of the substrate leads to tensile in-plane stresses in the top coat at high temperatures (assuming stress-free coating at room temperature). Stress relaxation in the top coat leads to compressive stresses in the top coat upon cooling the TBC sample. The top coat under compressive stress stores energy that leads to crack propagation (as the critical energy-release rate is exceeded) and finally to failure of the coating. For APS TBCs, the in-plane stress leads to edge delamination of the coating. Due to the sample edge, even if it is rounded, additional stresses occur at the edge of the sample causing coating delamination starting at the edge of the sample. As explained in the introduction, columnar TBCs will build-up less tensile stresses at elevated temperatures compared to APS TBCs. At the edges, the opening of the columns might even be easier, resulting in even lower stresses at these locations and hence, lower compressive stresses at room temperature. This might shift the failure location towards the radial center of the coating.

As a result of the reduced compressive stresses at the edge and on the whole sample while cooling the TBC, columnar TBCs are expected to provide less stored elastic energy for crack propagation compared to APS TBCs. This should lead to the higher thermal cycling performance of columnar TBCs compared to APS TBCs. As there are further influencing factors, for example oxidation, which is described in the following section, the positive effect of ODS bond coats on the performance of columnar TBC samples was significant, but the performance of the columnar samples in this study stayed below the performance of APS TBCs.

In Figure 2, images of the thermally cycled samples with conventional IN 0 and oxide-dispersion-strengthened bond coats ER 10 and IN 10 are shown. Conventional bond coat samples (IN 0 A,B) as well as oxide-dispersion-strengthened bond coated samples (ER 10, 30 A) showed a blue surface after delamination of the top coat in the center of the sample. The blue surface clearly indicates cobalt containing oxide growth, which contributed to the failure of the coating. The TBC systems using oxide-dispersion-strengthened bond coats showed minor buckling of the top coat (ER10 and 30 B, IN10 and 30). For both types of samples with both delamination and buckling, the top coat delaminates at the TGO (visible in Figure 3).

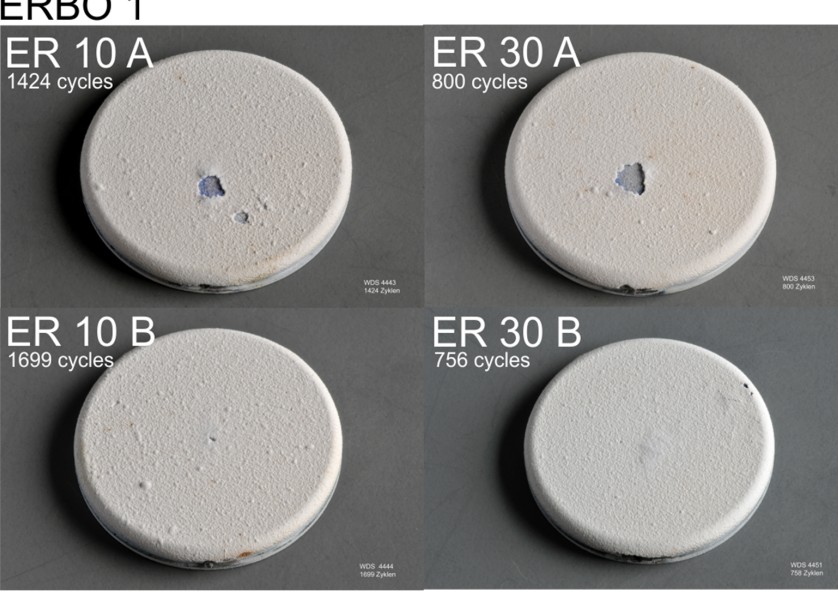

**Figure 2.** Top view of thermally cycled TBCs with yttria-stabilized zirconia SPS top coat after testing in the burner-rig facility for different substrate materials, ODS bond coats with (10% alumina) and (30% alumina) and conventional bond coats (0% alumina), A and B indicating two samples cycled under the same conditions. Sample specifications, see Table A1.

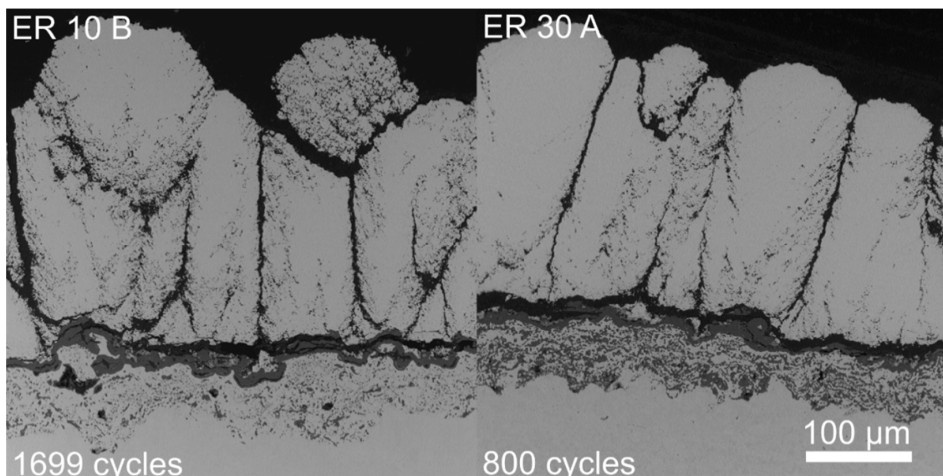

**Figure 3.** SEM cross section of thermally cycled TBCs using columnar yttria-stabilized zirconia SPS top coat, Co-based alumina oxide-dispersion-strengthened bond coats (10% and 30% alumina) and conventional Co-based bond coat underneath (on ERBO 1 substrate—not visible).

Even though the position of failure initiation in columnar (center) and APS TBCs (edge) was different for the specimen used in the burner-rig experiment, similar effects contributed to the failure of the coating. As described in the introduction, one effect causing failure of TBCs is significant oxide growth causing stresses and cracks at the interface. The second is crack propagation and coalescence, and subsequently delamination influenced by the energy-release rate of the TBC caused by the CTE mismatch between the substrate, the bond coat and the top coat. Reducing the CTE of the bond coat by the use of oxide-dispersion strengthening has a strong influence on the number of cycles to failure of columnar TBC systems.

Comparing the number of cycles to failure of columnar TBCs using oxide-dispersion-strengthened bond coats with TBCs using conventional bond coats, a clear performance increase is visible (Figure 4). The increase is possible even on rough interfaces (7.3–8.4 μm $R_a$), which is linked with the effect of the reduced CTE of the oxide-dispersion-strengthened bond coat.

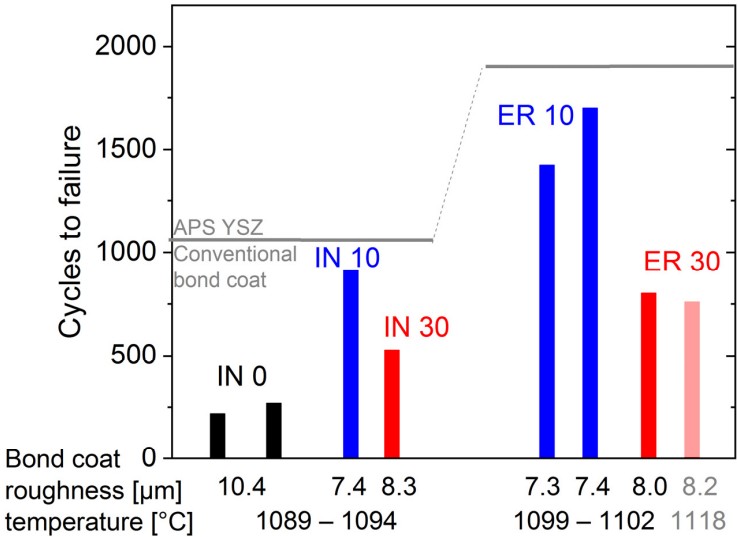

**Figure 4.** Number of cycles to failure of thermally cycled columnar TBCs with yttria-stabilized zirconia top coat using conventional bond coat (0% alumina) and oxide-dispersion-strengthened bond coats (10% alumina and 30% alumina) on two substrates: Inconel 738 (left) and ERBO 1 (right). The number of cycles to failure of APS TBCs at comparable bond coat temperatures were added by the grey horizontal lines (APS data taken from [2,16]).

As mentioned in the introduction, the number of cycles to failure of columnar TBCs on rough bond coat surfaces using conventional bond coats is rather low (IN 0, left columns in Figure 4) compared to APS TBCs with conventional bond coats (indicated by horizontal bars). As suggested in [11,24], the early failure might be due to the sensitivity of the columnar structure to cracks at the tips of the wavy top coat/bond coat interface caused by the CTE mismatch of top coat and bond coat. Here mainly local radial stress levels come into play [11,24]. The compressive stresses at the valleys of a rough bond coat/top coat interface during cooling of the TBC are reduced, which is related to the porosity surrounding the column (especially visible in Figure 5a) and is caused by the suspension-deposition process being affected by the rough bond coat surface [11,24]. As a result of the lower compressive-stress levels, columns can locally detach from the bond coat before significant oxide growth occurs (Figure 5b).

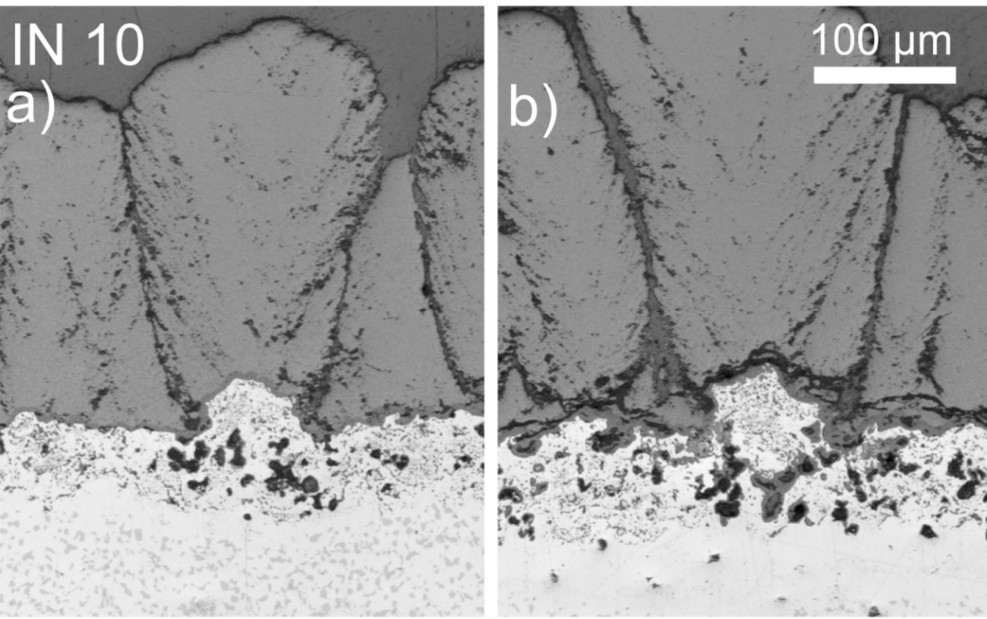

**Figure 5.** Laser-microscope images of TBC systems using columnar yttria-stabilized zirconia SPS top coat, Co-based alumina-oxide-dispersion-strengthened bond coat (10% alumina) and conventional bond coat (on Inconel 738 substrate–not visible) indicating porosity to the column attachment (**a**) and a single detached column (**b**).

The susceptibility of columnar TBCs to stresses at the bond coat/top coat interface is indicated by the different thermal cycling behavior of columnar and APS TBCs, as well as the influence of using ODS bond coats, which provide reduced stresses at the interface due to a lower CTE mismatch between the top coat and the bond coat. The potential of columnar TBCs using ODS bond coats is underpinned by clearly outperforming columnar TBCs with conventional bond coat, this is in line with the behavior of conventional APS TBCs [2]. The performance increase can be described by the multiplication factor of the number of cycles to failure by comparing two TBCs at the same bond coat temperature. The columnar TBC samples with 10% alumina ODS bond coats showed about a four-times-higher number of cycles to failure than columnar TBCs with conventional bond coats (IN10 to IN0 in Figure 4). The increase due to the ODS bond coats with 30% alumina for APS TBCs was lower, 1.5× to 1.7× for Inconel 738 [16] and ERBO 1 [2] substrate, respectively. As shown in the following analysis, the influence of oxidation and failure of columnar TBCs using 10% ODS leads to the conclusion that ODS bond coats combined with low-thickness APS and columnar TBCs lead to a two-fold increase in the number of cycles to failure compared to APS TBCs.

The oxide-scale thicknesses of the columnar TBCs (Figure 6) were comparable (10% ODS) or lower (30% ODS) than those of APS TBCs on samples from a previous study

(samples of [2] were analyzed with the same technique for comparison). The small scale thickness suggests minor stresses introduced by the oxide scale and therefore a minor influence on the detachment. Even the TBCs using 30% alumina oxide-dispersion-strengthened bond coats with different oxidation resistance [2] (IN30, ER30) showed thin TGOs and a high number of cycles to failure compared to conventional bond coats (Figures 4 and 6).

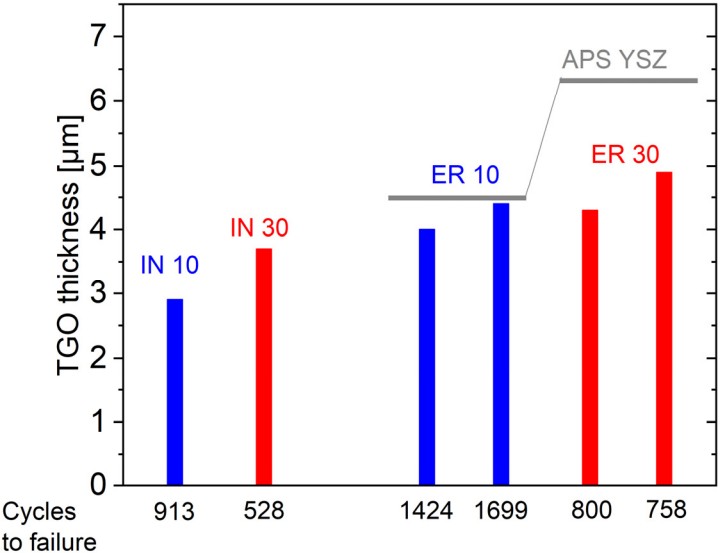

**Figure 6.** Thermally grown oxide (TGO) thicknesses of columnar TBCs using oxide-dispersion-strengthened bond coats (10% and 30% alumina) on two substrates Inconel 738 (left) and ERBO 1 (right), number of cycles to failure were added to the bars, TGO thicknesses of APS TBCs were added by horizontal bars.

Besides the significant effect of stresses at the bond coat/top coat interface on the thermal cycling performance of columnar TBCs, in-plane stresses in the coating are a lifetime-influencing factor. For columnar TBCs, these stresses should be reduced compared to APS TBCs, and a further reduction due to a lower coefficient of thermal expansion of the substrate material is also visible. The columnar TBCs on ERBO 1 therefore showed a higher numbers of cycles to failure (Figure 4) because ERBO 1 provided a lower CTE [16]. Some thermally cycled samples showed buckling of the top coat (Figure 2), which indicates some sufficient compressive stresses to drive delamination cracks even in columnar-structured top coats, while the integrity of the columnar layer was not destroyed. These findings are in line with the behavior of APS TBCs and indicate the presence of in-plane stresses in the coating, even in columnar-structured TBCs. A further study of this phenomenon is ongoing.

From these discussions, the following mechanisms can be established. At first, a detachment of small areas or single columns (Figure 5) are caused by stresses on the roughness scale due to the CTE mismatch of the bond coat and top coat. Afterwards, buckling of the top coat and detachment at the oxide scale is caused by the CTE mismatch between substrate and top coat (Figure 3). The second mechanism takes place after the first one, as TBC buckling needs a certain crack length at the interface. These findings are related to results of Trunova et al. and the modeling approach of Vaßen et al. [25], who described the crack length in APS TBCs as increasing significantly faster at a certain oxide-scale thickness [26,27]. Cernushi et al. also reported several small cracks at early stages of degradation [28].

## 4. Summary and Conclusions

This paper describes the effects of oxide-dispersion-strengthened bond coats on the performance of columnar-structured thermal barrier coatings. The results showed a sig-

nificant effect of the mismatch of the coefficient of thermal expansion between the bond coat and the top coat on the lifetime of columnar TBCs. Oxide-dispersion-strengthened bond coats provide a reduced CTE mismatch compared to conventional bond coats and thereby allow a significant performance increased by an up-to-four-times-higher number of cycles to failure. This could be attributed to the reduced stresses at the bond coat/top coat interface.

Some oxide-dispersion-strengthened TBC samples showed buckling of the top coat without macroscopic coating spallation or spallation on only small area (compared to APS TBCs).

The number of cycles to failure confirmed the impact of the different ODS bond coats on the performance of the suspension plasma sprayed columnar TBCs.

**Author Contributions:** Writing—original draft, C.V.; Writing—review & editing, D.E.M.; D.Z., O.G. and R.V. All authors have read and agreed to the published version of the manuscript.

**Funding:** This work was partly funded by SFB Transregio 103 (Project number B6).

**Institutional Review Board Statement:** Not applicable.

**Informed Consent Statement:** Not applicable.

**Data Availability Statement:** Not applicable.

**Acknowledgments:** We thank our cooperation partners for the supply of the single crystal superalloy material. The authors acknowledge the contribution of the following colleagues in our institute: Ralf Laufs, Frank Kurze and Karl-Heinz Rauwald for the invaluable assistance during plasma spraying and Martin Tandler for the effort with the cyclic burner-rig tests.

**Conflicts of Interest:** The authors declare no conflict of interest.

## Appendix A

**Table A1.** Sample specification "*" adapted from similar bond coated Inconel samples from [2,11].

| Sample | IN 10 | IN 30 | IN 0 A | IN 0 B | ER 10 A | ER10 B | ER 30 A | ER 30 B | Figure 1 Left | Figure 1 Right |
|---|---|---|---|---|---|---|---|---|---|---|
| Number cycles to failure | 913 | 528 | 218 [11] | 269 [11] | 1424 | 1699 | 800 | 758 | 2294 [2] | - [11] |
| Substrate | Inconel 738 | Inconel 738 | Inconel 738 | Inconel 738 | ERBO 1 | ERBO 1 | ERBO 1 | ERBO 1 | ERBO 1 | Inconel 738 |
| Bond coat | ODS 10% | ODS 30% | Con. 0% | Con. 0% | ODS 10% | ODS 10% | ODS 30% | ODS 30% | Con. 0% | Con. 0% |
| Bond coat temperature (°C) | 1093 | 1090 | 1094 | 1089 | 1099 | 1095 | 1102 | 1118 | 1085 | |
| Top coat thickness (µm) | 261 ± 20 | 303 ± 15 | 289 | 317 | 231 ± 13 | 275 ± 16 | 233 ± 10 | 221 ± 7 | 534 | |
| Bond coat: standard (µm) ODS (µm] | 147 ± 8 58 ± 9 | 150 ± 5 54 ± 7 | 240 - | 240 - | 129 ± 7 45 ± 7 | 134 ± 5 55 ± 6 | 131 ± 7 55 ± 6 | 139 ± 9 54 ± 6 | 238 - | |
| Bond coat roughness (µm) | 8.4 | 7.4 | 7.8 * | 7.8 * | 8.0 | 8.3 | 7.2 | 7.3 | 9.4 | |
| Beta depletion (µm) | 25 ± 12 | 41 ± 13 | | | 43 ± 19 | 86 ± 16 | 33 ± 7 | 101 ± 8 | 56 ± 3 | |
| TGO thickness (µm) | 2.9 ± 0.3 | 3.7 ± 0.7 | | | 4.0 ± 0.5 | 4.4 ± 0.5 | 4.3 ± 0.3 | 4.9 ± 0.7 | 7.5 ± 1 | |
| Crack density (1/mm) | 2.9 | 2.6 | | | 3.2 | 2.9 | 2.0 | 3.4 | - | |
| Column density (1/mm) | 7.2 | 7.8 | 10.1 | 10.1 | 7.0 | 7.1 | 9.3 | 7.2 | - | |
| Sample (internal) | 4442 | 4448 | 3882 | 3884 | 4443 | 4444 | 4453 | 4451 | 4310 | |

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
