# Peer review of "Effect of Low-CTE Oxide-Dispersion-Strengthened Bond Coats on Columnar-Structured YSZ Coatings"

_coatings, doi:10.3390/coatings12030396_

Round 1

Reviewer 1 Report

Dear Author,

  1. The SEM image of coating powder is not depicted and the explanation have to be explained.
  2. The XRD plot is not depicted for the coated samples and detailed explanation to be given
  3. The References has be further included.

Author Response

Dear Reviewer 1,

Thank you for reviewing our article, we have the following answers (bold letters) to your comments:

1. The SEM image of coating powder is not depicted and the explanation have to be explained.

- To not overcrowd the article the authors would not like to show the  SEM image of the ODS bond coat powders again. In the reference [2] powder cross sections and properties are presented. The link to the reference is stated at the beginning of the experimental part line 115, 116.  Bond coat and top coat powders are commercially available and are not presented, as those are industry products.

2. The XRD plot is not depicted for the coated samples and detailed explanation to be given

- XRD of the samples surface would commonly show YSZ afther thermal cycling the phase of the YSZ might have changed. These phase changes are common and known for this material in thermally cycled state thereby we would not like to add XRD

1. The References has be further included.

- Upon an additional check the refenrece list of the article is complete

The resubmitted article, with changes marked in red, underwent an additional spell check.

Sincerely,

Christoph Vorkötter and all co-authors

Reviewer 2 Report

  1. Can you add an element mapping of Fig.3 and Fig. 5? The reader does not know the element for coating layer materials
  2. Why not add a schematic for your method or research.

Author Response

Dear Reviewer 2,

Thank you for reviewing our article, we have the following answers (bold letters) to your comments:

1. Can you add an element mapping of Fig.3 and Fig. 5? The reader does not know the element for coating layer materials

- In figure 3/5 SEM/laser microscopy of the thermally cycles TBCs are shown. To increase readability the layer descriptions were included in the Figure annotations line (in detail: Co-Based bond coat, Alumina oxide dispersion strengthened bond coat, Yttria stabilized Zirconia top coat, Ni-based supperalloy) . The authors would not like to add an element mapping of the images, as this is hard efford to take, while the information is in the experimental part and is now included in the figure annotations.

2. Why not add a schematic for your method or research.

- Schematics of the work would include 1. High energy milling, 2. Sieving, 3. Thermal spray and thermal cycling of the samples. The authors are aware that this are several techniques used for this recearch, but like to state that there is nothing special about these techniques. As there is nothing special, we would not like to add scematic of processing. 

Changes in the resubmitted article are marked in red.

Sincerely,

Christoph Vorkötter and all co-authors

Reviewer 3 Report

Submitted manuscript presents results on investigation of effects of oxide dispersion strengthened bond coats on the performance of columnar structured thermal barrier coatings. It is shown that the lifetime of columnar TBCs is greatly depended on  the thermal expansion coefficient mismatch between bond coat and top coat, and oxide dispersion strengthened bond coats  reduce TEC  mismatch  and increase performance  by up to 4x higher number of cycles to failure. The article is well organized and structured, contains a sufficient amount of analyzed literature data, a clear description of the results obtained and further comparison with data from previously done studies. I recommend to accept the manuscript for publication. However, I have a few minor comments and suggestions.

Please, define APS acronym in the text.

Lines 33-34 "different thermal expansion coefficients of the top coat (α= 11-10-6 K-1)": the authors provide one TEC value. How does this value relate to "different coefficients". Or does the top coat always have only this value?

Perhaps that would be more convenient to use acronym for the "thermal expansion coefficient".

Author Response

Dear Reviewer 3,

Thank you for reviewing our article, we have the following answers (bold letters) to your comments:

Submitted manuscript presents results on investigation of effects of oxide dispersion strengthened bond coats on the performance of columnar structured thermal barrier coatings. It is shown that the lifetime of columnar TBCs is greatly depended on  the thermal expansion coefficient mismatch between bond coat and top coat, and oxide dispersion strengthened bond coats  reduce TEC  mismatch  and increase performance  by up to 4x higher number of cycles to failure. The article is well organized and structured, contains a sufficient amount of analyzed literature data, a clear description of the results obtained and further comparison with data from previously done studies. I recommend to accept the manuscript for publication. However, I have a few minor comments and suggestions.

1. Please, define APS acronym in the text.

- APS acronym is now defined at the beginning

2. Lines 33-34 "different thermal expansion coefficients of the top coat (α= 11-10-6 K-1)": the authors provide one TEC value. How does this value relate to "different coefficients". Or does the top coat always have only this value?

- the plural coefficients was used due to comparison of the top coat thermal expansion coefficient and the bond coat and component thermal expansion coefficient in the later part of the sentence. The top coat is made of "one" material yttria stabilized zirconia the corrseponding thermal expansion coefficient is (α= 11-10-6 K-1)

3. Perhaps that would be more convenient to use acronym for the "thermal expansion coefficient".

- by that we would need to change the title of the article. Additionally this wording would not be in line with the wording in previous publications. We would like to keep the wording and even shortenend the article by using CTE.

The resubmitted article, with changes marked in red, underwent an additional spell check.

Sincerely,

Christoph Vorkötter and all co-authors

Round 2

Reviewer 1 Report

The Paper gives an extensive work on coatings.